# Innovation of Women Farmers: A Technological Proposal for Mezcalilleras' Sustainability in Mexico, Based on Knowledge Management

**David Israel Contreras-Medina** [1,*], **Sergio Ernesto Medina-Cuéllar** [1], **Julia Sánchez-Gómez** [2] **and Carlos Mario Rodríguez-Peralta** [2]

1   Departamento de Arte y Empresa, División de Ingenierías Campus Irapuato-Salamanca DICIS, Universidad de Guanajuato, Carr. Salamanca-Valle de Santiago km 3.5+1.8, Comunidad de Palo Blanco, Salamanca 36885, Mexico; se.medina@ugto.mx

2   CONACyT-Centro de Investigación y Asistencia en Tecnología y Diseño del Estado de Jalisco A.C. CIATEJ, Av. de los Normalistas No. 800, Colinas de la Normal, Guadalajara 44270, Mexico; jsanchez@ciatej.mx (J.S.-G.); cperalta@ciatej.mx (C.M.R.-P.)

*   Correspondence: di.contreras@ugto.mx

**Abstract:** Currently, technology usage is a fundamental asset for creating, developing, and implementing innovations; however, these are not available to everyone, which is accentuated in women with agricultural occupations. The present study develops a proposal of technologies for mezcalilleras' sustainability from Oaxaca, Mexico, based on a knowledge management methodology, through the application of questionnaires in face-to-face sessions, field visits, and statistical analysis to explore the imbalances enclosed in the agave–mezcal activity seen as its problems, failures, and barriers, as well as its correlation with the identified technological routes. The results reveal that a technological platform creation, the fabrication of fiber optic refractometer, a metal roof construction, the design of a horizontal distiller–fractionator, the employment of metal containers and production of glass bottles, and the generation of a software and an application, are the suitable technologies, according to the mezcalilleras' requirements. This proposal can be important for academics, policymakers, and producers who wish to revitalize traditional knowledge of the small-scale sectors in Mexico through new ways of interaction with external agents and customers, new ways of production, and previous years' production analysis.

**Keywords:** mezcalilleras; technologies; agave-mezcal activity; SECI knowledge management model

## 1. Introduction

Technology, defined as knowledge, methods, and equipment such as computers and the internet [1,2], has played a key asset in human performance for creating, developing, and implementing innovations over time [3–5]. The creation of computing technology in the 1940s [6], the inclusion of technologies in corporations in the 1970s and 1980s [7], passing to the use of the internet in the early 1990s [8], have been some of the benefits of technology over the past century. From this, at the beginning of the 21st century, digital transformation of physical products into virtual assets became present [9,10]; advancing to the fifth generation of 5G mobile technologies [11] is another example of technological evolution that is changing the way that the social, organizational, and industrial sector works in recent times [12], although with inequalities in the use and benefits for the male and female population [6,13–15].

In the present-day, disproportion in the use of technologies is evident, since we have only reached almost 50% of the world's population through computers, smartphones, softwares, applications, and different platforms [16,17]; while the remaining little more than 50% still faces barriers and exclusion [18], denying them access to knowledge and personal

opportunities [19], and exacerbating social inequalities among countries, communities, and individuals, mainly women [20–23].

This situation is evidenced in many studies, such as [24], exposing the inequalities related to internet and mobile device usage and online interaction in citizens from 28 member states of the European Union (EU-28); [25] showed the digital division in vulnerable groups from Bolivia, Brazil, Dominican Republic, Ecuador, Guatemala, Finland, and Poland; [26] revealed the separation of digital tools in the poorest zones in Mexico, registering that internet access and digital skills are not available for everyone. This is further supported by the digital adoption index, in which the discrepancies are manifest in people located in nations classified as high income such as Hong Kong, in which the sub-index of people is established as the highest in the world with 0.91—on a 0–1 scale—which means that the local population is adopting technologies to expand their opportunities and improving their welfare, comparing for example with Mexico—classified as an upper-middle-income economy—with 0.43, located in 89 position of 367 economies, making evident a significant heterogeneity [27,28].

As the second-most populous country in Latin America, Mexico has a total population of 126,014,024 inhabitants, of which 51.5% are women, with economic participation of 45% [29,30]. This proportion does not mean better personal and professional opportunities for the female gender in the country, but quite the opposite, since Mexico faces social and economic inequalities and disadvantages, in addition to high discrimination against women. This is evidenced in the 77% of men doing primary, secondary, and tertiary economic activities against 45% of women, demonstrating an inequality of 32% [31–33].

The agriculture activity of the primary sector in Mexico, which is the backbone of the economy, contributing 58% of the total value of production [34], is the most significant opportunity for the employment of 7.6 million rural men and women and represents 42% of total income [35,36]. Due to this, agriculture is considered a fundamental activity for the local population and national reservoir of the country's culture; besides that, food production has its origin in this sector, almost entirely [37–40].

In January 2020, one of Mexico's traditional agricultural products, mezcal, was registered as one of five goods with excellent commercial value, having exportations of 148 million dollars, an increase of 8.28% [41]. However, most of the benefits were concentrated mainly in the industrial units, relegating, punishing, and marginalizing the small-scale producer to only the supply of inputs [42–44], although they suffer the most from the country's environmental, social, and economic problems [45].

The production of mezcal in Mexico has its home in Oaxaca, with 46.62% of agave cultivation and 90.1% of mezcal of the national output [46,47]. The mezcal has, in a cactus with rosettes, green leaves and, in the center, a tall with a flower on top, named agave plant, its primary raw material [48]. This plant has about 211 species worldwide, of which 159 are located in Mexico [49]. Its processing encloses an artisan system, sheltered by the experiences, beliefs, and traditional knowledge of rural peasants and indigenous peoples, for the proper utilization of natural resources [50–53], and the production of the so-called drink of the gods: mezcal [54].

In practice, agave–mezcal in Mexico is produced by mezcaleros masters or mezcalilleros—the small-scale male producers who own the pots and pans to transform the mature Agave plants into mezcal, using their traditional knowledge that is a thousand years old [55,56]—in a context of poverty, social exclusion, and reduced economic income of those who have inherited the tradition of its production. This activity is developed facing a set of barriers and difficulties in the organization and control of the process, the lack of articulation with markets, in addition to the absence of technology [57], causing contamination and the reduction of profits, forcing many to migrate [58–60]. This fact pushed the women, called mezcaleras masters or mezcalilleras—the woman who produces and likes mezcal—to care for agave cultivation and mezcal production in the present-day [61,62].

There are differences among mezcalilleros and mezcalilleras in the agave-mezcal process, due to their physical complexion. For instance, in the recollection of agave pineapples, due to its weight and diameter of 80 kg and 37 cm in average [63], or in the transfer by truck of these to the Palenque, because this activity was carried out by men [64].

Historically, the female gender is invisible and discriminated against daily in all areas with unequal treatment [65,66], this is also evident in the mezcalilleras' productive context since they have received poor attention and support [61,67]; therefore, there is a risk that, derived from this exclusion, this ancient activity, including its traditional knowledge, will be lost. Due to this, it is essential to promote the inclusion of women and detect their needs [68–71], based on their experiences and perspectives for a sustainable technological and systemic change [72–74].

## 2. Literature Review

As the soul of entire worldwide agri-food systems, traditional knowledge becomes a fundamental activity enclosed in productive chains, not only for food production but also to face the significant challenges in the field of sustainability; therefore, it is essential to preserve it and revitalize it, and technologies are necessary for this [75,76].

The conjunction between knowledge and technology may have had its origin in Schumpeter's economic theory. It was recorded that the innovation has evolved from widening to deepening, establishing the identification of technological opportunities, optimal conditions, and cumulative knowledge as a pattern-base [77]. The sense of knowledge has been the foundation in the evolution of the human being, since our first bipedal ancestor Lucy, in trying to adapt to the environment, until today, when it is considered the soul of innovation [78–80].

The knowledge in nature implies a process in a tacit–explicit sense, which goes from commitment, beliefs, and experiences of human reality to external assets, which, applied in an optimal time and space associated with the discovering of opportunities, generates new knowledge, innovations, and technologies [77,78,81–85]. This knowledge, combined with technologies, has been developed in research and practice studies [86,87], involving women located in productive food sectors worldwide to support decision making, build strategies, and create new management practices through digital tools. This is evident in [88], which evaluated the effect of knowledge and Information Technologies tools in supply chain management success. One study [89], showed the positive impact of knowledge on the innovation of women entrepreneurs. The research of [90] explored the significant relationship between knowledge and innovative performance in the service sector.

Additional studies such as [91] provided a gender experiment to improve new management practices of Ugandan women related to maize crops, sharing knowledge through informational videos which exposed an appropriate input use, and management practices using different strategies to obtain higher yields. Another study [92], explored the role WeChat played for 25 rural women from China, showing a variety of motivations for using this technological platform, such as promoting their traditional business through exhibiting their products, communicating with suppliers and customers, and learning strategies, signifying not only the opening of new learning and self-advancement oppurtunities, but also a tool for the construction of innovative female citizens. Further [93], described the traditional knowledge of 48 women combined with technical machinery, evidenced in manual activities. One example is the report about Doña Viviana, a Bolivian woman who expanded her knowledge through workshops for learning to elaborate new products using quinoa [94].

Studies have only pointed out the separation of the productive sector with technologies, including its lack of relationship with, and its propensity to reconnect it with, digital tools in Mexico. For example, the analysis of [95] to detect the relationship of the production chain indicates a rupture between twelve Zapotec peasant producers of mezcal from Oaxaca, Mexico, and modern technologies, causing the abandonment of this activity, through localized agri-food methodology besides participatory observation. The research of [60]

expressed that the traditional agave distillate system is technologically disadvantaged, and applied historical analysis. The investigation of [96] exposed a study on four indigenous women and five men from Union Hidalgo and Chicapa de Castro from Oaxaca, Mexico, related with the use of technologies such as cellphones, to make a series of informational videos for revitalizing traditional knowledge revealing the desire of reconnection between Zapotec people with technologies, through comunalidad methodology. The study of [97] channeled the information technologies for improving the programming of the local radio station of indigenous language, with a native community named Tseltal and local leaders in Chiapas, Mexico, to promote and strengthen indigenous language and community growth. It is worth mentioning that women, the Tsetsal population, represent 50.9% [98]. The research of [68], which explored the relationship between gender differences that influence the use of technologies in 24 Mayapan women from Yucatan, Mexico, found that women have better communication skills using cell phones utilizing SMS messaging, WhatsApp, social media, and internet interaction, using a focus group, observations, and an interview methodology. As well, [99] identified higher internet usage in rural female youth, searching for information and maintaining communication with family and friends, applying the econometric methodology.

The authors of this research acknowledge the contribution of previous studies in Mexico; however, the lack of complete analysis, in addition to the poverty statistics at the world level, reinforcing the sense that the investigations, have not been relevant for small producers [100], mainly for the female genre.

Several methodologies have been used in combining technology, traditional knowledge, and productive food context requirements, for example, participatory technology development and dissemination methods, under the stages of identifying, generating, testing, adapting, and promoting to help local difficulties [101]; the modern scientific knowledge systems proposal for integrating appropriate technologies [102]; the six-phase methodology to propose a digital model for the distribution of traditional knowledge [103]. However, knowledge management methodology, particularly Nonaka's model, involving commitments, beliefs, experiences, and knowledge, combined with the flexibility and interaction in the identification of expectations of technological routes of [104], is emerging as two complementary methodologies, since one has been the basis for the development and adaptation for traditional knowledge over time [56,105], as well as being the most used and cited [106,107]. The other is applied in detecting technologies based on perspectives, and, in addition to that, both have already been used in the context of Mexico [108–111].

In Mexico, the Instituto Federal de Telecomunicaciones concluded that local women utilize a more significant proportion digital technologies such as the mobile smart telephone, social networks, and internet in 51.6% in the country [112,113]. This sense is homologated towards rural areas [99] and, today, it was established at Information, Communication, and Technologies ICT, which are features of highest interest to develop actions in favor of women in the country [114]. For this, the research develops a technological proposal for mezcalilleras' sustainability from Oaxaca, Mexico, through four stages: (1) to identify mezcalilleras' requirements enclosed in the traditional knowledge of agave-mezcal activity in Mexico; (2) to describe the agave-mezcal production process of the mezcalilleras; (3) to explore technological routes; (4) to propose technologies for mezcalilleras' sustainability.

This research contributes to knowing mezcalilleras' traditional knowledge enclosed in the production of agave-mezcal; in illustrating the agave-mezcal productive process; in the design of innovation routes; in the proposition of technologies for mezcalilleras' sustainability from Oaxaca, Mexico, according to their requirements.

This study contains four sections. The first section, introduction, presents the history and evolution of digital technologies, their scope and gaps, and the status in the world and Mexico, also exposing the transcendence and problems of agriculture, specifically of agave-mezcal production, since the women's context. The second reveals the literature review, showing the conjunction between technologies and knowledge, displaying local and worldwide studies, involving the women located in productive food sectors, exposing

their propensity in the use of digital tools, actions, and methodologies used, and finishing in the objective and contribution on the present study. The third shows the materials and methods used, the justification of the sample and model used for data collection, the instrument, and the sequence of information analysis, reliability, and validity. The fourth presents the results recording the location and characteristics of the localities, agave-mezcal processes, technological resources, mezcalilleras' description, and the traditional knowledge enclosed in agave-mezcal production. This part also presents a statistical analysis to verify variables' correlations and the proposal of technologies for mezcalilleras' sustainability. The fifth displays the conclusions, implications, and recommendations for future studies.

## 3. Materials and Methods

The present study was carried out with 28 mezcalilleras, which signifies 36% of total agave-mezcal enterprises located in Oaxaca where a woman is in charge, representing a proportion of around 393,986 L produced, the origination of 1267 direct jobs, and 5789 indirect jobs in the state by year [115–117]. The groups were formed by 10 mezcalilleras from Miahuatlán de Porfirio Díaz, 9 from San Juan del Río, 3 from San Baltazar Chichicápam, 2 from San Luis Amatlán, 1 from San Pedro Teozacoalco, 1 from Santa Catarina Minas, 1 from San Miguel Suchixtepec, and 1 from Villa de Zaachila, Oaxaca, Mexico.

To identify mezcalilleras' traditional knowledge about ideas, information, and knowledge enclosed in the interaction of agave-mezcal activity in Mexico, questions based on the Socialization, Externalization, Combination, and Interiorization SECI layers of [81] we employed; in addition, technological routes, particularly considering the vision layer of [104], were designed and applied through face-to-face sessions and field visits. These were constructed following dimensions proposed by [106], adapted according to the scheme of [110], involved in non-profit organizations in Mexico, and homologated of the instrument of [108], applied in a small-scale agri-food Mexican context (see Figure 1).

For the Socialization layer, related with the interaction and sharing experiences [81], the following questions were asked:

(S1). As mezcalillera, do you share your knowledge with other people to produce mezcal?
(S2). As mezcalillera, do you build productive projects involving another mezcalillera?
(S3). As mezcalillera, do you get involved with another mezcalillera?
(S4). As mezcalillera, do you collaborate and communicate with external agents (government, institutions, and others)?

In Externalization, associated with the articulation of tacit knowledge into explicit forms [81]:

(E1). As mezcalillera, do you promote new ways of producing mezcal?
(E2). As mezcalillera, do you record your activities in formats that support the control of agave-mezcal production?
(E3). As mezcalillera, do you sell all your mezcal production?
(E4). As mezcalillera, do you consider the customer's needs for your production?

For Combination, linked with the dissemination of existing information [81]:

(C1). As mezcalillera, do you analyze the profitability before producing agave-mezcal?
(C2). As mezcalillera, do you analyze the feasibility of the implementation of projects?
(C3). As mezcalillera, do you analyze your internal capacities before producing agave-mezcal?
(C4). As mezcalillera, do you evaluate the quality of your production?

For Internalization, related with the embodying explicit knowledge into tacit knowledge [81]:

(I1). As mezcalillera, do you analyze the agave-mezcal production from previous years?
(I2). As mezcalillera, do you analyze the production of agave-mezcal from previous years with your group?
(I3). As mezcalillera, would you be interested in hiring personnel or acquiring equipment to improve your production?

(I4).　As mezcalillera, do you apply new ideas for the agave-mezcal production?

For technological routes:

As mezcalillera, what kind of equipment expectation would you require to improve the agave-mezcal activity?

All questions about SECI's model had a Likert five-type scale response options, composed from 5 to 1, wherein 5 = Very Interested (VI), 4 = Interested (I), 3 = Neither Interested Nor Disinterested (NIND), 2 = Little Interested (LI), 1 = Nothing Interested (NI). The responses were summed and divided against a total (448) to statistically normalize the process and identify the balance level in the four layers [118]. These were used for the screening imbalances (gaps) as problems, failures, or barriers (needs), comparing the frequencies of the highest level of each question versus balance level, according to the studies [108,110,119–121] used in the Mexican context. Regarding technological routes, equipment expectations were collected in an open question and were registered and ordered according to their frequency, following the recommendations of [109,111], applied in Mexico.

To the proposal, the current agave-mezcal process was described following the recommendations of [122], starting from the cultivation of agave to the sale of mezcal to the client, collecting the technological resources used, through field visits. After this, and based on technological routes, the criteria to evaluate and select the technologies recommended by [123,124] was applied, according to the need of the mezcalilleras, the opportunity of application, and to the availability of knowledge for its construction. Finally, statistical correlations between the imbalances such as the problems, failures, or barriers (needs) detected in the mezcalilleras' SECI process, with the equipment as technological routes, were executed, following [125,126].

The age, years producing, sale price as socio-demographic information, besides context-place of mezcalilleras, following the theory of [105,127], were included to evaluate the statistical correlation and practical development of technologies' proposal. These variables were already used in the coffee agriculture chain by [109,111].

It is worth mentioning that mezcalilleras participants were aware of the aim of the study and were asked if they wanted to participate, respecting the autonomy of the community to decide their internal methods of socialization, in agreement with [128].

*Analysis, Reliability, and Validity*

The responses of the Likert five-type scale related to the Socialization, Externalization, Combination, and Interiorization model were analyzed, recorded, ordered, and graphed by each question. as well, the most selected level (5 to 1) of each SECI layer was delineated to make visual mezcalilleras' activity for detecting imbalances as problems, failures, or barriers (needs).

In the same sense, the current version of the agave-mezcal process was described and mapped. The age, years producing, sale price, and equipment expectation were statistically analyzed and matched with mezcalilleras' SECI model to propose technological routes, using the software IBM SPSS© Statistics v21. The context-place was registered and analyzed to check technologies' practical implementation. Each technology proposition was filtered through criteria to evaluate it and select it, as in [123,124], and it was drawn using Microsoft Visio Professional© 2021.

Originally, SECI's instrument used in this study was developed and evaluated by Contreras-Medina, D.I. in the research of [108], following the Cronbach alpha test to check reliability level and the expert's opinions for validity [129,130], applying it in the agave-mezcal context for Mezcaleros' producers, which gives validity and reliability to its application with mezcalilleras, following the recommendations of [131], about selecting an existing instrument.

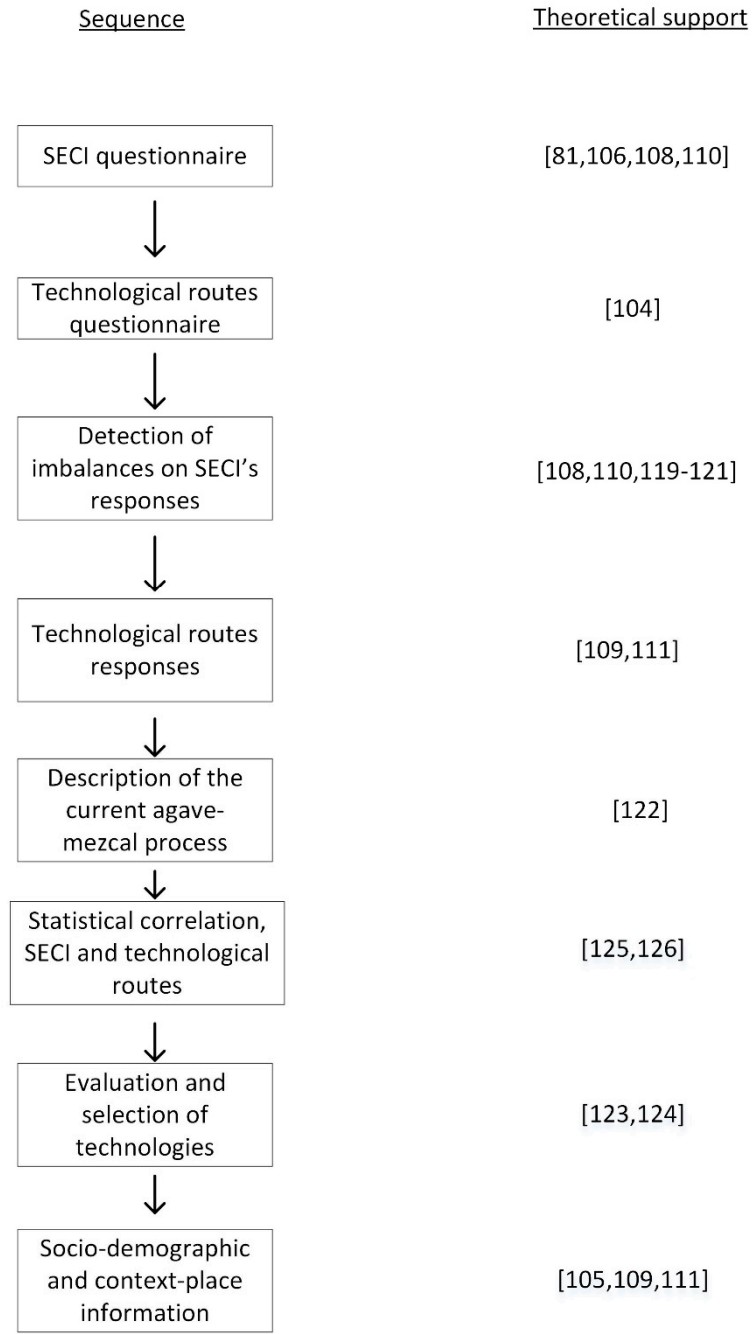

**Figure 1.** Flowchart of methodologies applied to the study.

## 4. Results and Discussion

### 4.1. Context-Place of Mezcalilleras

The mezcalilleras' context-place in Oaxaca, Mexico registers heterogeneous social and economic variables that limit the development of its local population. This is the case of the mezcalilleras from the localities of Miahuatlán de Porfirio Díaz, San Juan del Río, San Baltazar Chichicápam, San Luis Amatlán, San Pedro Teozacoalco, Santa Catarina Minas, San Miguel Suchixtepec, and Villa de Zaachila, incorporated in the study.

Currently, Miahuatlán de Porfirio Díaz registers 50,375 inhabitants, of which around 76% live in poverty and 34% in extreme poverty. The 59% of residents of those 15 years-old and older have an elementary school education, 86% of 25 years-old and older know to read and write, 98% of houses have electricity service, 24% have a computer, 30% have internet access, 84% count with a mobile cell phone, and 13% have a landline phone. San Juan del Rio records 1372 inhabitants, of which 84% are estimated to live in poverty and 39% in extreme poverty. Seventy-nine percent of the local population of 15 or more years have an elementary school education, 82% of those 25 years old and older can read and write, 99% of homes have electricity service, 10% have a computer, 11% have internet access, 34% possess a mobile cell phone, and 54% have a landline phone. San Baltazar Chichicápam registers 2576 inhabitants, of which around 74% live in poverty and 30% in extreme poverty. The 53% of residents of 15 years old and older have an elementary school education, 74% of those 25 years old and older know to read and write, 98% of houses have electricity service, 8% have a computer, 19% have internet access, 73% possess a mobile cell phone, and 1% have landline phone. San Luis Amatlán records 3829 inhabitants, of which 88% live in poverty and 49% in extreme poverty, approximately. The 68% of the local population of those 15 years old and older have an elementary school education, while the 72% of 25 years old and older can read and write, 97% of homes count on electricity service, 5% have a computer, 15% have internet access, 58% possess a mobile cell phone, and 28% have a landline phone. San Pedro Teozacoalco registers 1153 inhabitants, of which around 78% live in poverty and 39% in extreme poverty. The 69% of residents of 15 and more years have an elementary school education, 85% know to read and write, 96% of houses have electricity service, 4% have a computer, 10% have internet access, 31% have a mobile cell phone, and 5% a landline phone. Santa Catarina Minas records 2067 inhabitants, of which around 73% live in poverty and 31% in extreme poverty. The 69% of the local population of those 15 years old and older have an elementary school education, 86% of those 25 years old and older can read and write, 99% of homes have electricity service, 13% have a computer, 42% have internet access, 86% have a mobile cell phone, and 6% have a landline phone. San Miguel Suchixtepec registers 2932 inhabitants, of whom 83% live in poverty and approximately 49% in extreme poverty. The 59% of residents of 15 and more years have an elementary school, 79% know to read and write, 98% of houses have electricity service, 15% have a computer, 44% have internet access, 65% have a mobile cell phone, and 17% have a a landline phone. Villa de Zaachila records 46,464 inhabitants, of which around 76% live in poverty and 25% in extreme poverty. The 54% of the local population of 15 years old and older have an elementary school education, 92% of 25 years old and older can read and write, 96% of homes have electricity service, 25% have a computer, 29% have internet access, 91% have a mobile cell phone, and 8% have a landline phone [132,133]. (see Figure 2, Table 1). The economic situation of poverty and extreme poverty is homologated not only to the mezcalilleras and mezcalilleros by having the earnings of mezcal as a subsistence method, obtaining around 7.5% profit on cost per liter, using for daily spending, but also, to all states in Mexico, mainly those located in the south-southeast of the country such as Oaxaca, Guerrero, and Chiapas, since they register up to three times and in some up to 20% compared to the entities of the north [134–136]. A similar situation of poverty is found in the Population of Africa and South Asian countries such Zambia, Guinea, or Pakistan with 54.4%, 43.7%, and 24.3%, however, the standard of living has been transformed with the introduction and monitoring of the use of technology [137,138].

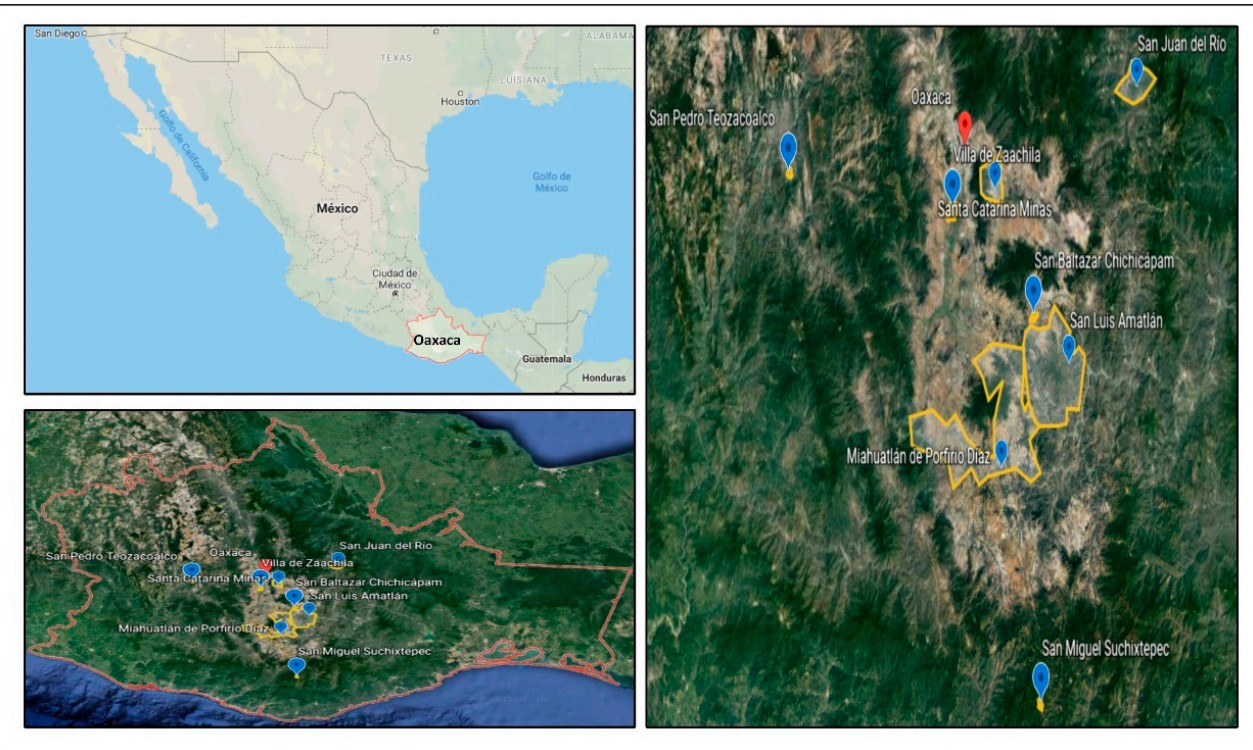

**Figure 2.** Location of Miahuatlán de Porfirio Díaz, San Juan del Río, San Baltazar Chichicápam, San Luis Amatlán, San Pedro Teozacoalco, Santa Catarina Minas, San Miguel Suchixtepec, and Villa de Zaachila, in Oaxaca, Mexico.

**Table 1.** Context-place variables percentages in the function of the total population by locality.

| Variables/Locality | Miahuatlán de Porfirio Díaz | San Juan del Río | San Baltazar Chichicápam | San Luis Amatlán | San Pedro Teozacoalco | Santa Catarina Minas | San Miguel Suchixtepec | Villa de Zaachila |
|---|---|---|---|---|---|---|---|---|
| Total population | 50,375 | 1372 | 2576 | 3829 | 1153 | 2067 | 2932 | 46,464 |
| Poverty | 76% | 84% | 74% | 88% | 78% | 73% | 83% | 76% |
| Extreme poverty | 34% | 39% | 30% | 49% | 39% | 31% | 49% | 25% |
| Education level | 59% | 79% | 53% | 68% | 69% | 69% | 59% | 54% |
| Literacy | 86% | 82% | 74% | 72% | 85% | 86% | 79% | 92% |
| Electricity service | 98% | 99% | 98% | 97% | 96% | 99% | 98% | 96% |
| Computer availability | 24% | 10% | 8% | 5% | 4% | 13% | 15% | 25% |
| Internet access | 30% | 11% | 19% | 15% | 10% | 42% | 44% | 29% |
| Mobile cell phone | 84% | 34% | 73% | 58% | 31% | 86% | 65% | 91% |
| Landline phone | 13% | 54% | 1% | 28% | 5% | 6% | 17% | 8% |

The age and antiquity in the mezcal production of mezcalilleras reveal an average of 44 years old and 27 producing mezcal. In this sense, the sale price of mezcal is established around 300 Mexican pesos (US 14.6 dollars) per litter. At the same time, the equipment expectations focus on infrastructure for traditional factories (called Palenque), specifically for the measurement of sugar in agave, for the roof of the oven, and for the alembic (alambique in Spanish), as well as in the packaging and sales for improving agave-mezcal production. Additionally, there is a need for association (see Table 2). These results expose a younger average age of mezcalilleras located in Oaxaca, compared to that reported by [139], registering an average of 67 years old of mezcalilleras from Guerrero, Mexico.

**Table 2.** Mezcalilleras' description.

| Mezcalilleras' Description | Characteristics |
|---|---|
| Age | 44 years old |
| Years producing | 27 years |
| Price per litter | 300 (US 14.6 dls). |
| Equipment expectations | Measurement of sugar in agave |
| | Roof of the oven |
| | Alembic |
| | Packaging |
| | Sales |
| | Association |

Regarding the years producing mezcal, it is reflected that the average age at which knowledge transfer begins is 17 years old, establishing a relative similarity with that reported by [140], in which the starting age of 18 years is handled only in male producers. About the price, the range established by mezcalilleras is a little lower in a proportion of 413 Mexican pesos (US 20.28 dollars), compared with that reported by [117] for a bottle of 750 mL. Regarding equipment expectations, the requirement to improve the process is mainly measuring sugar in agave, the roof of the oven, alembic, packaging, sales, and the association. This is in line with the study of [141], in which the need to innovate the production process of mezcal and commercialization to improve its competitiveness was recorded.

The most used variety for the production of mezcal by mezcalilleras is espadin. The production process starts with agave selection, followed by the cutting of agave of root and leaves called "jima" with a shovel, for later collection in a wheelbarrow, and placement of the agave pineapples or heads in a truck, to transport them to the Palenque which is the place where mezcal is produced. In Palenque, agave pineapples are baked in a stone room or a hole dug in the ground for two to five days, grinding the cooked pineapple and extracting its juice through the "trapiche", which is a metal wheel pulled by a horse. The liquid is placed in wooden fermentation vats for two to four days, depending on climate, to later go to the first distillation through "alambique", a clay pot where the juice is heated to turn it into steam through a copper tube. The next step is to remove the ethanol to continue with the second distillation and give the mezcal the degrees of alcohol allowed by the applicable regulation, storage, packaging, and sale by unit or block (see Figure 3, Table 3). This process is, in general, in line with the expose by [50], in which the sequence of harvesting and cutting agave, cooking, mashing, or milling (grind), fermentation, first and second distillation, and bottling, was registered.

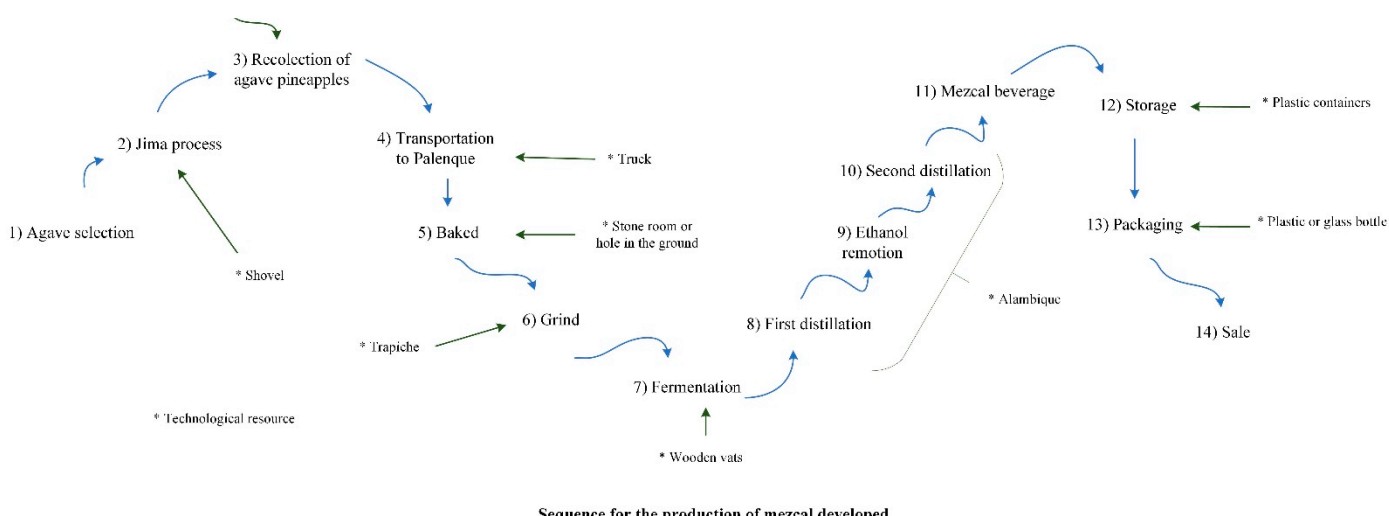

**Figure 3.** Current production process of agave-mezcal from the mezcalilleras.

**Table 3.** Technological resources are currently used in the mezcal process.

| Activity | Technological Resources |
| --- | --- |
| Jima | Shovel |
| Recolection | Wheelbarrow |
| Transportation | Truck |
| Baked | Stone room/hole in the ground |
| Grind | Trapiche |
| Fermentation | Wooden vats |
| Distillation (first and second) | Alambique |
| Ethanol remotion | |
| Storage | Plastic containers |
| Packaging | Plastic or glass bottle |
| Sale | Glass bottle |

The traditional knowledge about ideas, information, and knowledge, enclosed in the interaction of agave-mezcal activity developed by mezcalilleras, reveals that balance was founded at Interested level I, reflecting significant heterogeneity in almost all SECI's layers. For example, in the Socialization layer, related to interaction and sharing experiences, the results expose a selection in all Likert five-type scales of NI, LI, NIND, I, and VI. This same sense was found for Externalization associated with the articulation of tacit knowledge into explicit forms, Combination linked with the dissemination of existing information, and Internalization related with embodying explicit knowledge into tacit knowledge [81], manifesting a significant variability (see Figure 4). The Likert five-type scale most selected was I, reflecting a balance in this level, and the Socialization as the most balanced layer; however, the preponderance presented in the S4 in NIND level revealed an imbalance (gap) and a problem, fault, or barrier [119–121], while S1 and S2 in I, and S3 in VI, were the majority of the answers chosen by the mezcalilleras. Internalization is classified as the second least balanced layer in this line since it is registered to I1 and I2 in LI as the most chosen level, exposing an inequity, and I3 and I4 in I. Externalization, placed in the third position, shows E1 and E2 in NIND level and E4 in NI as preponderance level, displaying a disproportion, while E3 is located in VI. Combination has C1 in NI, C2 in LI, C3 in NIND as dominance levels, presenting the highest discrepancy, and only C4 is in I level (see Figure 5). These results are in line with the results exposed by [108], revealing imbalances in all SECI layers, contrary to the established Socialization layer which is the most imbalanced and combination as the least, showing in male producers (mezcaleros) (a similar sequence is registered in [108]).

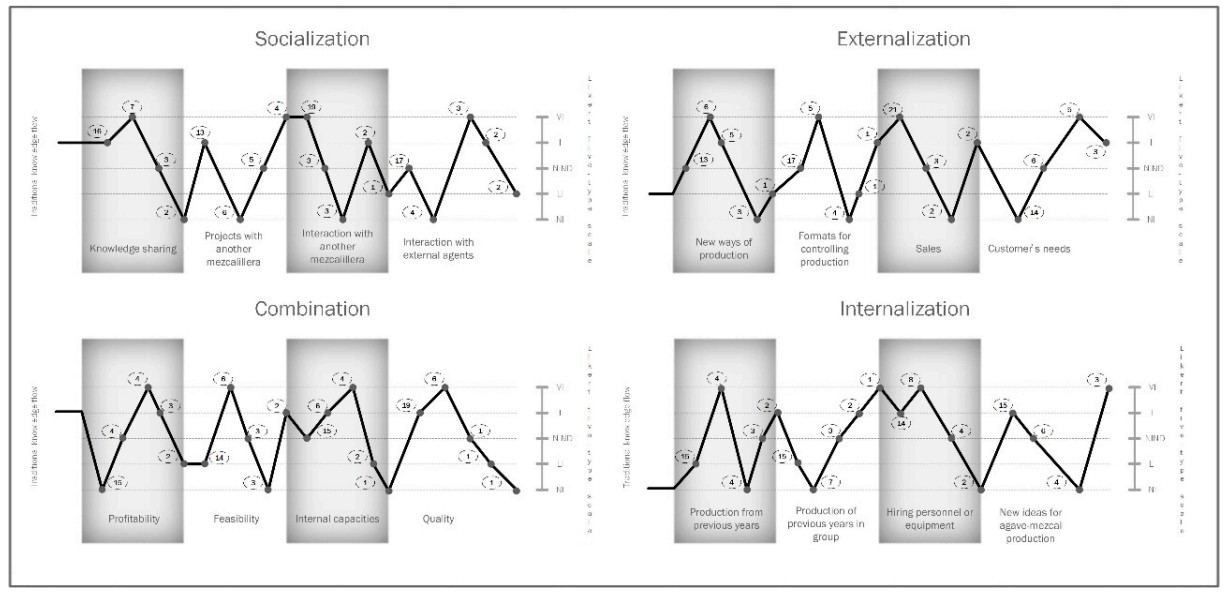

**Figure 4.** Mezcalilleras' traditional knowledge dynamics enclosed in the agave-mezcal production.

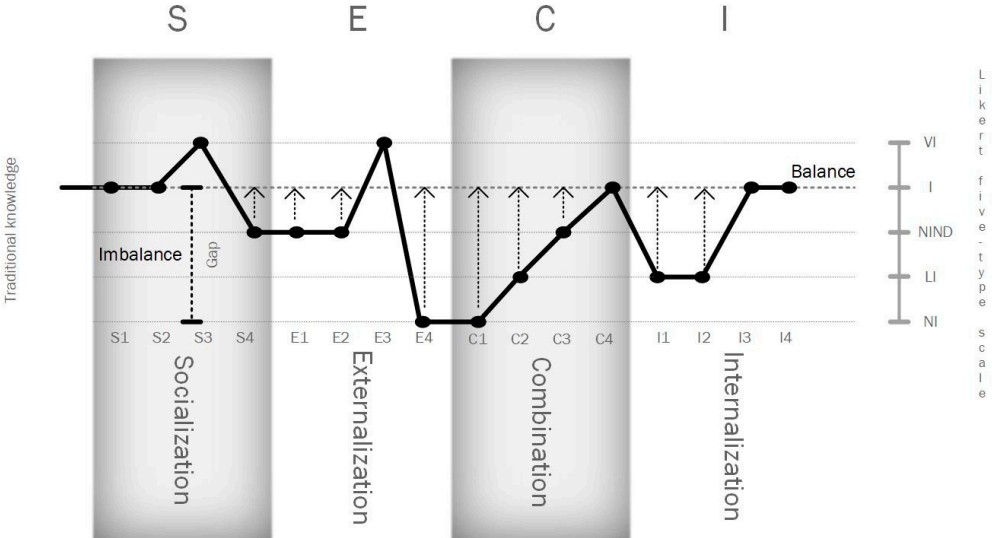

**Figure 5.** Mezcalilleras' traditional knowledge dynamics imbalances enclosed in the agave-mezcal production.

*4.2. Statistical Analysis*

The Shapiro–Wilk statistical test was performed based on the sample size to evaluate the normality of the SECI's layers [125]. The result determines ($p > 0.05$), representing a normal distribution. In this sense, the variables of age, years producing, price, and equipment requirements against SECI layers results were evaluated through the Pearson correlation test ($1 - r + 1$) to check the correlation and dependence of each one [126]. The results were registered as follows from the highest to lowest:

Age with Socialization S4r = 0.222, S1r = 0.128, S3r = 0.082, and for S2r = −0.192; with Externalization E4r = 0.468, E1r = 0.224, E3r = 0.222 and E2r = −0.580; with Combination C1r = 0.196, C2r = −0.186, C4r = −0.341 and C3r = −0.450; with Internalization I3r = 0.099, I1r = −0.201, I2r = −0.209, and I4r = −0.369.

Years producing with Socialization S4r = 0.392, S3r = 0.169, S1r = −0.009 and S2r = −0.249; with Externalization E4r = 0.180, E3r = −0.123, E1r = −0.322 and E2r = −0.469; with Combination C1r = 0.072, C2r= −0.533, C4r= −0.671 and C3r = −0.674; with Internalization I3r = 0.137, I2r = −0.279, I4r = −0.343, and I1r = −0.528.

Price with Socialization S1r = −0.006, S3r = −0.294, S4r = −0.328 and S2r = −0.390; with Externalization E2r = 0.265, E3r = 0.160, E1r = −0.210 and E4r = −0.667; with Combination C1r = 0.129, C2r = 0.098, C3r = −0.534 and C4r = −0.551; with Internalization I1r = 0.163, I3r = 0.089, I2r = −0.358, and I4r = −0.415.

Equipment with Socialization S4r = 0.625, S3r = 0.505, S1r = 0.237 and S2r = 0.209; with Externalization E4r = 0.329, E1r = 0.161, E3r = −0.207 and E2r = −0.447; with Combination C3r = −0.222, C4r = −0.324, C2r = −0.578 and C1r = −0.589; with Internalization I2r = 0.148, I4r = −0.022, I3r = −0.463, and I1r = −0.717 (see Table 4).

**Table 4.** SECI and mezcalilleras activity statistical correlation.

| SECI/Mezcalilleras Description | Age | Years Producing | Price Per Litter | Equipment |
|---|---|---|---|---|
| Socialization | | | | |
| S1 | 0.128 | −0.009 | −0.006 | 0.237 |
| S2 | −0.192 | −0.249 | −0.390 | 0.209 |
| S3 | 0.082 | 0.169 | −0.294 | 0.505 |
| S4 | 0.222 | 0.392 | −0.328 | 0.625 |
| Externalization | | | | |
| E1 | 0.224 | −0.322 | −0.210 | 0.161 |

**Table 4.** *Cont.*

| SECI/Mezcalilleras Description | Age | Years Producing | Price Per Litter | Equipment |
|---|---|---|---|---|
| E2 | −0.580 | −0.469 | 0.265 | −0.445 |
| E3 | 0.222 | −0.123 | 0.160 | −0.207 |
| E4 | 0.468 | 0.180 | −0.667 | 0.329 |
| Combination | | | | |
| C1 | 0.196 | 0.072 | 0.129 | −0.589 |
| C2 | −0.186 | −0.533 | 0.098 | −0.578 |
| C3 | −0.450 | −0.674 | −0.534 | −0.222 |
| C4 | −0.341 | −0.671 | −0.551 | −0.324 |
| Internalization | | | | |
| I1 | −0.201 | −0.528 | 0.163 | −0.717 |
| I2 | −0.209 | −0.279 | −0.358 | 0.148 |
| I3 | 0.099 | 0.137 | 0.089 | −0.463 |
| I4 | −0.369 | −0.343 | −0.415 | −0.022 |

For the above, the interaction with external agents (S4), knowledge sharing (S1), and the interaction with another mezcalillera (S3) of the Socialization layer presented a correlation with age. This sense is also present for new ways of production (E1), sales (E3), the consideration of customer needs (E4), profitability (C1), and the interest to hire personnel or acquire equipment for improving agave-mezcal production (I3) of Externalization and Interiorization. At the same time, for the rest, there is no correlation. This means that the imbalances registered in S4, E1, and E4 present a dependency with age, therefore, taking into account this socio-demographic variable can support overcoming the problems, barriers, and difficulties of these points. These results are in line with the study of [95], in which the age of producers, mainly the youngest (45 years old on average), seek new ways to produce the mezcal and to articulate with customers and external agents, to reduce unfavorable effects of a socioeconomic and productive context.

Regarding external agents (S4), and the interaction with another mezcalillera (S3) of Socialization, of Exteriorization customer needs (E4), the profitability analysis (C1) of combination, and the interest to hire personnel or acquiring equipment for improving agave-mezcal production (I3) of Interiorization, registered a dependency with the years producing agave-mezcal, whereas, with the rest, there is no correlation. This means that the imbalances registered in S4, E4, and C1 showed a dependency with the years producing, therefore its consideration and attention could resolve these elements' problems, barriers, and difficulties. The results are in line with the study [142], which claims that profitability is linked to association with external agents and that it implicitly depends on the years producing because the majority is between 60 and 80 years old, limiting the entry of new knowledge and technologies [140].

Concerning the record of activities in formats (E2) and sales (E3) of Exteriorization, the feasibility of implementation projects (C2) of combination, besides the analysis of previous years (I1), and the interest to hire personnel or acquiring equipment for improving agave-mezcal production (I3) of Interiorization, exposed a dependency with the price. At the same time, for the rest, there is no correlation. This means that the imbalances of E2, C2, and I1 indicated a dependence on the price, therefore its protection can support resolving the problems, barriers, and difficulties to balance these components. This result is in line with the study of [143], in which it is stated that, since some time ago, mezcaleros do not have instruments for recording and analyzing their activities for cost evaluation, nor the organization for the incorporation of productive projects.

Concerning external agents (S4), the interaction with another mezcalillera (S3), knowledge sharing (S1), and productive projects involving another mezcalillera (S2) of Socialization, of Exteriorization customer needs (E4), and the new ways of production (E1), in addition to the analysis of the agave-mezcal production in the group (I2) of Interiorization, displayed a dependency with the equipment; at the same time, for the rest, there is no

correlation. This means that the imbalances of S4, E1, E4, and I2 revealed a dependence on the equipment, therefore its consideration and supply can resolve the problems, barriers, and difficulties to balance these components. These results are in line with the study of [144], in which it is recorded that the equipment is low-tech in its production, in addition to the fact that the interaction inside and outside is reduced due to the lack of access to technologies [145].

*4.3. Technological Proposal for Mezcalilleras' Sustainability*

Based on the correlation between the problems, difficulties, or barriers (imbalances) enclosed in the agave-mezcal process registered for Socialization regarding the interaction and sharing experiences specifically in the interaction with external agents (S4), for Externalization associated with the articulation of tacit knowledge into explicit forms particularly with new ways of production (E1), and the customer needs (E4), and for Internalization related with embodying explicit knowledge into tacit knowledge exactly with the production of previous years in the group (I2), with equipment expectations as technological routes, the proposal of digital technologies is as follows:

1. Against the problems, difficulties, or barriers in the interaction with external agents (S4), correlating the equipment expectation as an association, the proposition is creating mezcalilleras' digital platform as an adequate technology, combining hardware, software, and internet services into a digital platform [146]. This technology is the most effective means to promote equity and sustainable development [147], particularly of mezcalilleras through the association between external agents as government institutions, the academic community, non-profit organizations, and all people related to the agriculture sector, that want to interact with the mezcalilleras from Mexico, sharing support programs, events, research studies, service provision, sustainable practices, and all products that might add value to the agave-mezcal process. The proposal of the technological platform is carried out based on the availability of knowledge of a Research Center belonging to the National Council for Science and Technology in Mexico (CONACYT), located in Jalisco, Mexico [148], taking advantage of its experience in the design of these technologies. The relevance of this technology has been validated by the European Commission and U.S. Department of Agriculture (USDA) when creating a women platform called Eurogender (EIGE), which is a hub of online cooperation for advancing gender equality in Europe and beyond [149], and with the USDA Women in agriculture mentoring network, to connect women in agriculture and share their experiences [150].

2. Touching the problems, difficulties, or barriers in the new ways of production (E1), and associating the expectation for the measurement of sugar in agave, the consideration of a fiber optic refractometer as an easy and fast way to determine the sugar level of agave of mezcalilleras, combining the sensors for measurement, an algorithm for calculation, embedded in a low-cost portable exemplar, since the knowledge of another Research Center, located in Aguascalientes, Mexico, can be exploited [151]. This hand-model instrument could improve the sugar level, quality, and the exact time for the agave Jima, for increasing mezcal production, enhancing economic benefits, and to avoid cutting the plant at the wrong times and thereby to achieve economic and environmental sustainability, since this technology is already considered by the Organization for Economic Co-operation and Development (OECD) as a way to determine the quality of fruit and vegetables [152].

3. Continuing with problems of E1, relating the expectation in the oven roof, the installation of a metal rooftop, placed above the oven, so that water is emitted as a shower, similar to a gas scrubber, could be the option attending to mezcalilleras' demand. With this, the mezcalilleras could construct a sustainable process because they are protected from inclement weather and are trapping part of the pollutants produced by the combustion of firewood through the water. This technology is a part of a project developed by another Research Center located in Jalisco, Mexico to fire

bricks using specialized software to improve energy efficiency. This knowledge could be homologated for the process of agave-mezcal [153], since there are experimental designs for this purpose [154].

4. Remaining with problems of E1, now linking the expectation in the alembic (alambique in Spanish), the installation of a horizontal distiller-fractionator with hydraulic closure, valves, and thermometers for the passage of mezcal, adding a pot of the same material to reduce the boiling time and achieving environmental sustainability, could be a technological option for improving the control of the process. This type of technology has already been studied by an Educational Institution located in Jalisco, Mexico, experimenting in Mexican citrus; it could also be homologated to the mezcal process once extraction efficiency is improved and standardized [155].

5. Continuing with problems of E1, correlating the expectation in packaging, the installation of metal containers in the storage stage as the priority added to the production of glass bottles, taking advantage of the stone room for baking agave heads, could be a suitable and relevant technology for the sustainability of the process. This option recommends replacing plastic containers with metal for the quality conservation of mezcal beverage, besides recycling the glass, using essential tools of transfer, molding, and resting in the process, which has already been implemented in the sector [156]. This proposal could be developed taking advantage of the knowledge of various Research Centers located in the northern region of Mexico and those immersed in the context of mezcal production [157–159].

6. Again, addressing problems of E1 associated with the expectation in sales, a similar technology to (S4) could be followed. For this, creating a digital platform in which the mezcal produced by the mezcalilleras could be the most appropriate option for improving the link between producers with the market, reducing costs, and achieving social sustainability [160]. This strategy is to digitize the drink to offer it in Mexico and the world, implemented in different world regions [161,162]. This proposal could be developed through the knowledge generated by the National Council for Science and Technology in Mexico (CONACYT) and Educational Institutions in the north region [163,164].

7. Concerning the problems of knowing the customer needs (E1), taking into account the association's expectations would establish technology direction under the creation of software as the most pertinent in the present for improving the quality of the beverage. This software works once installed on a computer under an interface similar to that of a menu, in which the customer would register their opinion and considerer their preferences about the drink to cut only the wanted number and select variety and thereby achieve economic and environmental sustainability. This digital tool is applied to interact with customers quickly and easily. It is already considered by the Organization for Economic Co-operation and Development (OECD) as a part of the digital transformation [165], and it could be constructed by taking advantage of Educational Institutions' knowledge in Mexico City, developing it through free access software, improving producer–client interaction, and connecting with international organizations working in the country [166,167].

8. Relating to the problems about analyzing the production of previous years in the group (I2), correlating the expectation of association, the design of a technological application APP could be a suitable technological tool for improving decision-making. This app could include the variables of production by growing season, month or day, variety of agave used, price, and customer to which it was sold, registering through a mobile cell phone, providing real-time information, recording the latest production levels from previous years for being analyzed, and giving the plant a break to regenerate the fields and begin to achieve economic and environmental sustainability. The importance for controlling the production and using applications was already recognized by the Food and Agriculture Organization of the United Nations when promoting projects on the development of apps with an emphasis on

females, such as the mezcalilleras, for using information through digital technologies for increasing agricultural productivity [124,168]. Besides, the evaluation of yields and household women incomes importance has been recognized by World Bank and United Nations, and in promoting and developing projects in the context of poverty in the world to balance the role of women [169,170], and for empowering women farmers to grow their yield and incomes [171].

This app could be designed utilizing the knowledge of Mexican Institutions, using the last tendencies and technologies in the context of agave-mezcal activity [172,173].

The digital importance of tools such as platforms, apps, and software are being implemented through projects in different parts of the world focused on females. For this, it could well be implemented in the mezcalilleras' activities in Mexico, since they count with the context-place for implementing these technologies, starting with San Miguel Suchixtepec with the highest level of internet access. Technologies as fiber optic refractometer, metal rooftop, horizontal distiller-fractionator, metal containers, and glass bottles could be implemented in all localities since they only require training (see Figure 6).

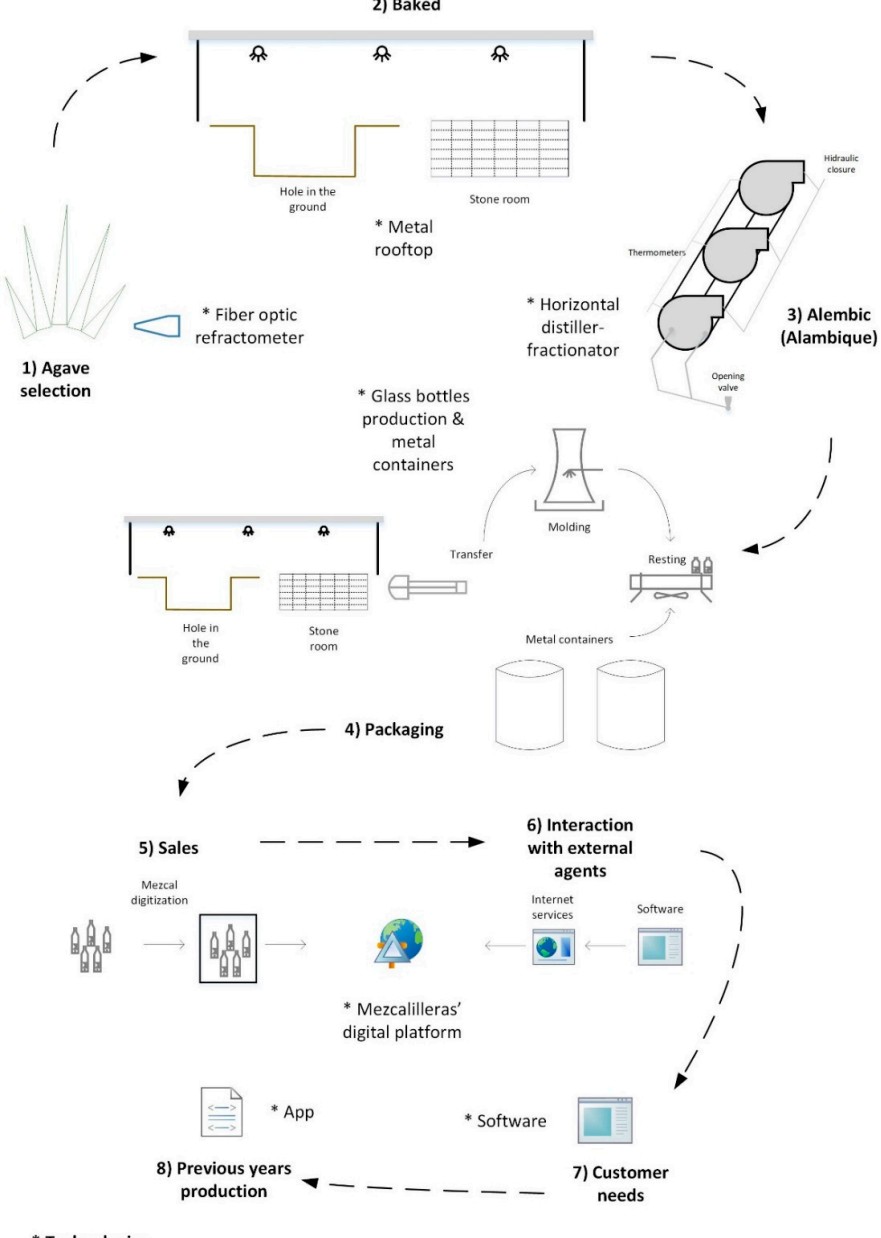

**Figure 6.** Technological proposal.

For the above, and because the mezcalilleras are the women-in-charge of mezcal production, local female producers could use this technological proposition. This affirmation is made in light of the findings of [174], in which it is stated that the adoption of technologies is stimulated when women have control over resources. Invariably, the implementation of a digital platform, fiber optic refractometer, metal rooftop, horizontal distiller-fractionator, production of glass bottles, software, and app, must be accompanied by experts for teaching mezcalilleras how to use digital technologies, following [175] recommendations, and requiring making explicit the benefits to avoid skepticism [176], in addition to complementing the support with government, academic, or non-profit institutions, or the related population.

## 5. Conclusions

Based on the objective of proposing digital technologies for mezcalilleras' sustainability from Mexico, based on knowledge management, this study reveals significant requirements (imbalances) enclosed in traditional knowledge of agave-mezcal production, determining the combination layer with the highest gaps, indicating the most crucial problem or failure. For the Externalization, Internalization, and Socialization, the findings expose minor imbalance and fewer problems.

Combination, as the most imbalanced layer, indicates problems related to profitability (C1), feasibility (C2), and internal capacities (C3). For the case of Externalization, Internalization, and Socialization, the inequities (problems) focus on the interaction with external agents (S4), new ways of production (E1), formats for controlling production (E2), customer needs (E4), production of previous years (I1), and the production of previous years in the group (I2). These imbalances manifest a correlation with age, years producing, and price per litter, while the highest is of equipment with S4.

Regarding the imbalances correlations with equipment, for technological routes, the results reflect an association with S4, E1, E4, and I2, therefore the mezcalilleras expectations as an association, measurement of sugar in agave, oven roof, alembic, packaging, and sales, could have been attended through technological proposition and have a positive influence for the interaction with external agents, new ways of production, to know the customer needs, and to record the production of previous years in the group.

A large proportion of equipment expectations are focused on the production process, which is correlated with the imbalance of new forms of production. Therefore, the technologies proposals, such as the fiber optic refractometer, metal rooftop, horizontal distiller-fractionator, metal containers, glass bottles, and digital platform, could solve the problems of new ways of production, in addition to software, the creation of an application to remove the barriers and interact with external agents, and customers to record the production of previous years. It is essential to mention that digital platform technology could also be used for local interaction to bring traditional knowledge among mezcalilleras and with men and those of the new generation of Oaxaca, promoting new management practices and different ways to use this technological tool, or the elaboration of new products, since all localities have internet access, and mobile cell phone and computer availability, the proposal could be implemented taking advantage of the fact that women are more likely to use Technologies, leaving intact agave-mezcal traditional activity

The age reflects a significant correlation in all layers, followed by the years producing and price, mainly in Socialization and Externalization, in which the most important is found. For this, the mezcalilleras of 44-year-olds must be prioritized for the entry of technologies proposal, followed by the antiquity in the production of mezcal of 27 years.

Based on the context-place variables of localities, the mezcalilleras' digital workspace technological platform and app proposal could be installed in San Miguel Suchixtepec, because it registers the highest internet access. However, technologies such as fiber optic refractometer, metal rooftop, horizontal distiller-fractionator, metal containers, and glass bottles could be implemented in all localities since they only require training.

Technologies proposed in this study should be interpreted with caution because of the lack of statistical representativeness. It is necessary to carry out a broader analysis

that considers more locations in this and other traditional chains in Mexico and the world, despite their contribution to mezcal production in Oaxaca, Mexico.

This study can be important for practitioners, academics, policymakers, and small producers, for improving mezcalilleras' sustainability, and trying to preserve and revitalize women's traditional knowledge.

**Author Contributions:** Conceptualization, D.I.C.-M.; methodology, D.I.C.-M., S.E.M.-C., J.S.-G. and C.M.R.-P.; software, D.I.C.-M. and S.E.M.-C.; validation, D.I.C.-M., S.E.M.-C., J.S.-G. and C.M.R.-P.; formal analysis, D.I.C.-M., S.E.M.-C., J.S.-G. and C.M.R.-P.; investigation, D.I.C.-M., S.E.M.-C., J.S.-G. and C.M.R.-P.; resources, D.I.C.-M., J.S.-G. and C.M.R.-P.; data curation, D.I.C.-M., S.E.M.-C.; writing—original draft preparation, D.I.C.-M., S.E.M.-C., J.S.-G. and C.M.R.-P.; writing—review and editing, D.I.C.-M., S.E.M.-C., J.S.-G. and C.M.R.-P.; visualization, D.I.C.-M., S.E.M.-C., J.S.-G. and C.M.R.-P.; project administration, D.I.C.-M., J.S.-G. and C.M.R.-P.; funding acquisition, D.I.C.-M., J.S.-G. and C.M.R.-P. All authors have read and agreed to the published version of the manuscript.

**Funding:** This research was funded by the National Council for Science and Technology of Mexico (Consejo Nacional de Ciencia y Tecnología CONACYT acronym in Spanish), under the project titled: Learning, training and analysis of the environment to strengthen value chains based on work with female mezcal masters in Oaxaca and Guerrero, with ID 06520.

**Institutional Review Board Statement:** The study was proposed and conducted according to the guidelines of the Declaration of Helsinki, asking if they want to participate, respecting their autonomy to decide their internal methods of socialization in agreement with [128], protecting the identity, approved by the National Council for Science and Technology of Mexico CONACYT, and developed by CIATEJ through technical manager of the project "Learning, training and analysis of the environment to strengthen value chains based on work with female mezcal masters in Oaxaca and Guerrero, with ID 06520, in 2018.

**Informed Consent Statement:** Informed consent was obtained from all subjects involved in the study.

**Data Availability Statement:** The information of this study can be consulted in its entirety within the project "Learning, training and analysis of the environment to strengthen value chains based on work with female mezcal masters in Oaxaca and Guerrero, with ID 06520". To know the information access requirements please see the page https://conacyt.mx/transparencia/ (accessed on 17 December 2020).

**Conflicts of Interest:** The authors declare no conflict of interest.

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
