# Peer review of "Innovation of Women Farmers: A Technological Proposal for Mezcalilleras’ Sustainability in Mexico, Based on Knowledge Management"

_sustainability, doi:10.3390/su132111706_

Round 1

Reviewer 1 Report

Overall, a very interesting manuscript. The subject matter is topical given the ongoing expansion of Mexico's distilled spirit sector and the potential for small-scale producers to participate more fully in this growth. The methodology appears to be sound and properly applied. To showcase this work, the authors would benefit from making improvements to the literature review and making the exposition of the methodology and findings easier to understand.

Literature review (in both the Introduction and Literature Review sections)

The literature review needs to be improved. The authors need to make sure that the works cited actually relate to the subject at hand. In a few places, the authors make assertions that are difficult to substantiate, and the authors would be better off just stating the facts.  For instance, on page 2, what makes agriculture the "backbone" of the Mexican economy? Is it surprising that food production has its origins in the agricultural sector?

Later on page 2, is it really true that small-scale mezcal producers "suffer the most from the country's environmental, social, and economic problems?" Uribe Reyes doesn't mention the mezcal sector at all in the work that the authors cite (42). Neither does the press release from IICA (39) nor the article in Comercio Exterior (41), and the thesis by Sanchez Juarez appears to be about coffee growers. But the authors cite all of these works to make claims about the mezcal sector.

On page 3, is the female gender really "invisible and discriminated against in all areas"? I am able to look online and easily find articles about women in the mezcal sector, so they don't seem invisible. Also, what is the connection of the article on honor killings and Islam to the mezcal sector?

Missing from the literature review is an overview of the mezcal sector and the major trends and issues facing the sector. Of particular importance are any differences among producers, especially along the lines of gender and scale of operation.

Context-place

The authors establish that the mezcal producers are located in communities with extreme poverty. What is the economic status of these producers, and how does it compare with the communities at large? How much do the producers earn from mezcal, and how much from other income sources?

Page 13: "Statistical", not "Statisticall." "Pearson" test, not "Person."

Exposition

Because some readers will not be familiar with the methodology, it is imperative to communicate the methodology, results, and findings in a clear fashion. At times, the findings get lost in technical terms (socialization, externalization, etc.). On pages 5-6, the questions could be numbered at the front of each question using the terms in parentheses (e.g., E1, E2, E3...) so that it is not necessary to have the term in parentheses at the end of the question as well.

Pages 12-14 are particularly dense, and many readers would benefit from additional text that guides them through this passage. For instance, why is it important to rank the Pearson test results? Is it necessary to report the rankings when a table with the test results follows? How could the text that follows the table be used to explain the importance of years of experience to a mezcal business and of transferring knowledge from one generation to the next, sometimes from one gender to another?

Finally, the conclusions in their current form emphasize technological improvements to complement traditional production techniques, with the topic of gender only mentioned once. The manuscript would have a much stronger ending if it summarized how digital platforms and intergenerational communication could be used to impart knowledge to producers, including women and the new generation.

Author Response

We appreciate your comments. All were valuable to reinforce the manuscript. ¡Thank you!

Reviewer 2 Report

The paper is pretty well-written. The authors can consider shortening the first few paragraphs in the introduction section that introduce technology backgrounds that is irrelevant to agriculture. One more round of English checks is preferred.

The authors have made great efforts in surveying women farmers' needs in mezcalilera's sustainability in Mexico. In response to their surveys, they proposed new technologies possible to help women farmers. The research fills the gap that women's needs in agriculture is usually neglected.

The authors have made lots of efforts to provide a detailed introduction, literature review, and surveying methods.

The main concern is that the writing is too lengthy, the English writing style is too oral. There are lots of long sentences such as line 21-26 in the abstract section, line 144-147 in the literature review section. I suggest the authors checking the paper again and making the paper shorter and more concise.

Some other suggestions include:

1) shorten the introduction section, especially those irrelevant to Mexican agriculture.

2) The last sentence of the literature review should belong to the introduction section.

3) There are too many percentages in the results and discussion sections.

4) Table 1 needs captions to explain the meaning of percentages.

5) Avoid complicated result descriptions and abbreviations in the conclusion section. The Conclusion section needs to be shortened as well.

6) Some of the conclusion sections can serve as a limitation subsection in the discussion section.

Author Response

(The authors gave the same response as above.)
